# Knowledge, attitudes, and practices toward Mpox among laboratory professionals in Zambia: A cross-sectional study

David Chisompola[1]*, John Nzobokela[2], Elijah Chinyante[2], Nanjela Chidima[2], Allen Chipipa[2], Charlotte Nyirenda[2], Edward Phiri[2], Lucky Kalyapu[3], Sepiso K. Masenga[4,5]*

1 Pathology Laboratory Department, Arthur Davison Children's Hospital, Ndola, Zambia, 2 Pathology Laboratory Department, Ndola Teaching Hospital, Ndola, Zambia, 3 Biomedical Science Department, Ndola College of Biomedical Sciences, Ndola, Zambia, 4 Division of Integrated Sciences, Livingstone Center for Prevention and Translational Science, Livingstone, Zambia, 5 HAND Research Group, School of Medicine and Health Sciences, Mulungushi University, Livingstone Campus, Livingstone, Zambia

* d.chisompola@gmail.com (DC); sepisomasenga@lcpts.org (SKM)

## Abstract

### Introduction

Monkeypox (Mpox), caused by the Mpox virus, is an emerging zoonotic disease of global public health significance. In Zambia, limited data exist on laboratory professionals' (LPs) preparedness to manage and prevent Mpox outbreaks. This study aimed to assess knowledge, attitudes, and practices (KAP) regarding Mpox among LPs in Zambia.

### Methods

A cross-sectional online survey was conducted among 293 LPs across Zambia from April to August 2025. A structured, validated questionnaire assessed socio-demographic data and KAP toward Mpox. Descriptive statistics, chi-square tests, and logistic regression analyses were performed using Stata version 15.

### Results

Of the 293 respondents, 52.2% demonstrated good knowledge, 47.4% had a positive attitude, and 7.5% exhibited good Mpox-related practices. Significant knowledge gaps were observed, including limited awareness of Mpox transmission (62.5%), complications (76.1%). Additionally, 72% of LPs expressed willingness or support for mandatory Mpox vaccination. LPs working in health centres/clinics (AOR = 0.07; 95% CI: 0.009–0.54; p = 0.11), private hospitals (AOR = 0.05; 95% CI: 0.003–0.74; p = 0.030), public hospitals (AOR = 0.06; 95% CI: 0.011–0.39; p = 0.003), and research or academic institutions (AOR = 0.10; 95% CI: 0.01–0.77; p = 0.027) were significantly

**Data availability statement:** The raw data underlying the results presented in the study have been uploaded as a Supporting information file.

**Funding:** The author(s) received no specific funding for this work.

**Competing interests:** The authors have declared that no competing interests exist.

less likely to report good Mpox-related practices compared to those working in district or provincial health offices.

## Conclusion

The study highlights a low knowledge, a relatively low positive attitude, and poor practices toward Mpox among Zambian LPs. Identified gaps underscore the important need for targeted education and capacity-building initiatives to enhance Mpox preparedness and response.

## Introduction

Monkeypox (Mpox) is a viral zoonotic disease caused by the Mpox virus, a member of the *Orthopoxvirus* genus in the *Poxviridae* family, which also includes variola (smallpox), cowpox, and vaccinia viruses [1,2]. Human infections occur through animal spillover or direct contact with lesions, body fluids, or respiratory droplets of an infected person, presenting with fever, lymphadenopathy, and a progressive rash that can lead to complications such as secondary infections, sepsis, or corneal scarring [3]. Historically, Clade I Mpox infections have been linked to a higher case fatality rate, ranging from 1.4% to 11%, compared to Clade II Mpox infections [4]. The 2022 global outbreak underscored its potential for human-to-human spread via close contact, necessitating rapid diagnosis, isolation, and vaccination for containment [5,6].

Globally, notable outbreaks have been reported to include the 2003 U.S. outbreak linked to imported prairie dogs [7], as well as travel-associated cases in the UK and Israel in 2018 [8,9]. As of November 2023, over 91,000 cases and 166 deaths had been reported across 116 countries [10]. The African continent has borne the heaviest burden, with the Africa CDC reporting a 160% increase in cases and a 19% rise in deaths, predominantly in the DRC [11]. By 2024, Africa had recorded over 14,000 cases and 524 fatalities, surpassing previous years' figures, with new outbreaks emerging in Burundi, Kenya, Rwanda, and Uganda [12]. Mpox has been endemic to the rainforest regions of Central and West Africa, with sporadic outbreaks outside the continent linked to travel or animal trade [1,3]. The global health importance of Mpox was highlighted when the World Health Organization (WHO) declared it a Public Health Emergency of International Concern (PHEIC) on August 14, 2024 [13]. This decision followed a surge in cases in the Democratic Republic of the Congo (DRC) and the virus's spread to previously unaffected African countries, placing neighbouring Zambia at heightened risk. If left unaddressed, Mpox can lead to severe complications such as secondary bacterial infections, respiratory distress, encephalitis, and, in some cases, death [14].

Zambia reported its first Mpox case in October 2024 [15], signalling an urgent need to evaluate healthcare professionals' (HCPs) such as laboratory professionals' (LPs) preparedness in managing potential outbreaks. HCPs play a critical role in outbreak containment and are regarded as being at elevated risk of acquiring severe infectious diseases while caring for patients [16]. Recent studies conducted across

different countries have highlighted that HCPs generally exhibit low levels of knowledge about Mpox, along with a lack of positive attitudes and suboptimal practices related to its prevention and management [17–19]. Yet there is limited data on their knowledge, attitudes, and practices (KAP) regarding Mpox in Zambia. Addressing this gap is essential for tailoring effective public health interventions, improving health education, and strengthening outbreak response strategies.

This study aimed to assess KAP related to Mpox among Zambian healthcare workers, identifying key gaps in awareness and behavioural practices. The findings will inform targeted measures to enhance Zambia's capacity to mitigate the spread and impact of Mpox.

## Methods

### Study settings and participants

We conducted a cross-sectional descriptive study in Zambia using an open online self-administered survey designed with Google Forms (see Table 2). This study was carried out from 29th April to 24th May 2025. A combination of convenience and simplified snowball sampling was employed to recruit participants. LPs were invited through emails and social media platforms such as WhatsApp, Facebook and LinkedIn. The study was conducted and reported in accordance with the Strengthening the Reporting of Observational Studies in Epidemiology (STROBE) [20], checklist to guide reporting, supplementary file S3 File. The study was a nationwide survey conducted in Zambia, targeting LPs from a wide range of facilities and practice settings.

The principal investigator and co-investigators of the study initiated recruitment through simplified snowball sampling [21]. This was done beginning with conveniently selected seed participants representing diverse healthcare professions, workplace settings, and geographic regions. To mitigate recruitment bias inherent in chain-referral methods, we implemented three safeguards: (1) limiting each participant to referring a maximum of three colleagues to prevent network clustering, (2) verifying professional credentials of all referred participants to maintain sample quality, and (3) supplementing with various professional groups to reach HCPs beyond referral networks. This approach balanced the efficiency of snowball sampling with intentional diversity preservation across key demographic and professional strata, while the referral cap and verification process specifically addressed potential overrepresentation of certain subgroups that could skew results.

### Inclusion and exclusion criteria

We included participants who were LPs and residents of Zambia at the time of the study, aged 18 years or older, able to understand and read the English language used in the questionnaire, provided informed consent to participate, and had access to the internet to complete the online questionnaire. Participants were excluded if they were under 18 years of age, not residing in Zambia, unable to provide informed consent due to cognitive or communication impairments.

### Sample size determination

The sample size was calculated using the sample proportion formula provided by the OpenAI online statistical tool. We employed an estimated prevalence of Mpox-related knowledge among HCPs, which is reported to be 26% [17], the confidence level was set at 95%, and the margin of error, set at 0.05. Under these assumptions, the minimum required sample size to ensure statistical robustness was calculated to be at least 296 participants. A total of 293 participants were recruited.

### Data collection instrument and data collection procedures

The data collection tool used in this study was a structured questionnaire adopted from validated instruments in previous studies [20,22,23], modified and developed using Google Forms. The questionnaire was designed in English and included essential components such as an introduction to the study, a confidentiality statement, and a clear indication that

participation was voluntary. It comprised four main sections: (1) socio-demographic information, including gender, age, education level, years of professional experience, type of institution and region of residence; (2) knowledge-based questions; (3) attitude-based questions; and (4) practice-based questions related to Mpox.

Prior to the main data collection, a pilot study was conducted with ten healthcare professionals to assess the content reliability and clarity of the questionnaire. Following this, the finalized survey was disseminated through WhatsApp Messenger, Facebook, LinkedIn, and email, with periodic reminders sent to enhance response rates. The Google Form was configured to accept only one response per participant using its built-in "limit to one response" feature, which prevents duplicate responses without collecting identifying information. No email addresses or identifying information were collected, ensuring participant anonymity. Participants were able to access and complete the questionnaire using either a computer or mobile device. The dependent variable in this study was the participants' level of knowledge, attitude and practices regarding Mpox. The independent variables included various sociodemographic characteristics such as age group, gender, level of education, healthcare professional, years of experience, current working setting, and Province of residence.

The data collection form had four sections with a total of 42 questions (Supplementary form S1 File). The first section consisted of nine (9) questions which addressed participants' sociodemographic and descriptive characteristics. These included age group, gender, level of education, healthcare professional, years of experience, current working setting, region of residence, province of practice and the name of the institution.

Each section of the questionnaire was scored to enable quantitative assessment. For the knowledge section, which included 12 questions, each correct answer was awarded one point, while incorrect or "not sure" responses received zero points. The total possible score ranged from 0 to 12. Based on the modified Bloom's cut-off point [24], the total score for each question, respondents were categorized into two groups: those with "good knowledge" scoring ≥80%, and those with "poor knowledge" scoring <80% Similarly, In the attitude section, the total attitude score ranged from 0 to 10. Attitude levels were categorized into two groups: "positive attitude" for scores ≥90% and "negative attitude" for scores <90%. In addition to the knowledge and attitude section, the preventive practice section, had a total practice score ranging from 0 to 10. Scores were categorized into two levels: "good practice" for scores ≥75% and "poor practice" for scores <75% [24]. The overall knowledge score was interpreted according to Bloom's cut-off point as high (80–100%), moderate (60–79%), and low (≤59%) [25].

## Data analysis

The dataset was first cleaned, tabulated, and assessed for consistency and completeness using Microsoft Excel, and subsequently analysed using Stata version 15. Data collected during the pilot study were excluded from the final analysis to preserve the validity and reliability of the results. The Shapiro-Wilk test ($P < 0.05$) was conducted to assess whether the data were normally distributed. Both descriptive and inferential statistical analyses were performed. Descriptive analysis covered demographic characteristics and responses related to healthcare professionals' KAP regarding Mpox. Categorical variables were summarized using frequencies and percentages. Inferential statistical analyses were conducted to assess associations between KAP levels and selected sociodemographic variables such as sex, profession, level of education, and workplace setting. Chi-square tests ($\chi^2$) were used to evaluate the relationship between categorical variables, such as good and poor knowledge and practices and positive and negative attitude. Logistic regression was used for multivariate analysis to determine factors associated with good knowledge, positive attitudes, and good practices related to Monkeypox. Statistical significance was set at $p < 0.05$.

## Ethical approval

This study was conducted in accordance with the ethical principles outlined in the Declaration of Helsinki. All participants were fully informed about the purpose, objectives, and scope of the study and provided digital written informed consent prior to participation. Participation was entirely voluntary, and all responses were collected anonymously to

ensure confidentiality. Ethical approval for the study was granted by the Mulungushi University School of Medicine Research Committee and the National Health Research Authority on April 22nd and 29th, 2025, under reference numbers SMHS-MU2-2025-10 and NHRA8653/21/04/2025.

## Results

### Basic characteristics of the participants

A total of 293 participants completed the questionnaire. The majority were male (n = 158, 53.9%) and aged between 25 and 34 years (n = 184, 62.8%). Most respondents resided in urban areas (n = 219, 74.7%) and held a diploma qualification (n = 133, 45.4%). Regarding work experience, the majority of participants (n = 113, 38.6%) had 6–10 years of experience. Additionally, most participants were employed in public health hospitals (n = 194, 66.2%) and were primarily based in the Copperbelt Province (n = 115, 39.2%). The full demographic characteristics are presented in Table 1.

### Knowledge, attitude and practices of laboratory professionals about human Monkeypox

Among the survey respondents, 52.2% demonstrated low knowledge of Mpox (using Bloom's cut-off point of ≤59%), 47.4% exhibited a positive attitude, and 7.5% reported good preventive practices (Table 2). Despite a low level of knowledge, significant knowledge gaps were observed: 62.5% were unaware of Mpox transmission modes, 76.1% did not know its potential complications, 53.9% were unsure whether an effective vaccine exists, and 36.5% could not identify the groups at higher risk of severe Mpox disease (Table 2).

### Factors associated with practices towards Monkeypox

Age, years of experience, and workplace setting were included in both univariate and multivariate logistic regression analyses due to their statistical significance (p < 0.05). In the multivariate analysis for age, laboratory professionals (LPs) aged 25–34 years (AOR = 0.95; 95% CI: 0.10–9.27; p = 0.971) and 35–44 years (AOR = 1.01; 95% CI: 0.19–5.40; p = 0.985) demonstrated no significant differences in Mpox-related practices compared to those aged >45 years. Similarly, for years of experience, LPs with 1–5 years (AOR = 1.02; 95% CI: 0.10–9.86; p = 0.981), 6–10 years (AOR = 0.46; 95% CI: 0.04–4.88; p = 0.522), and more than 11 years (AOR = 1.42; 95% CI: 0.08–23.4; p = 0.804) did not differ significantly in their Mpox-related practices compared to those with less than 1 year of experience. In contrast, workplace setting was a significant predictor. LPs working in health centres/clinics (AOR = 0.07; 95% CI: 0.009–0.54; p = 0.11), private hospitals (AOR = 0.05; 95% CI: 0.003–0.74; p = 0.030), public hospitals (AOR = 0.06; 95% CI: 0.011–0.39; p = 0.003), and research or academic institutions (AOR = 0.10; 95% CI: 0.01–0.77; p = 0.027) were significantly less likely to report good Mpox-related practices compared to those working in district or provincial health offices (Table 3).

## Discussion

To our knowledge, this is the first study to investigate the knowledge, attitudes, and practices of LPs regarding monkeypox in Zambia, thereby providing a foundation for evidence-based management strategies and policy formulation. This study assessed the KAP of 293 LPs and identified key factors influencing their responses toward monkeypox. Our findings reveal critical gaps in preparedness among Zambian laboratory professionals. 52.2% of respondents showed good knowledge, only 47.4% reported a positive attitude, and 7.5% reported good preventive practices.

Our study finding showed an overall knowledge score of 52.2%, which is classified as low according to Bloom's cut-off point. This was consistent with the 50% reported in Vietnam [22], 55.3% in Egypt [26], and 55% in Saudi Arabia [27]. The low knowledge score likely reflects a lack of targeted, continuing education on emerging infectious diseases like Mpox for laboratory professionals in Zambia. It may also indicate a reliance on informal information sources which may not be comprehensive or accurate. In contrast, other studies reported lower levels of knowledge compared to our findings, with

**Table 1. Characteristics of the study population.**

| Variable | Frequency (%) | Knowledge Score N (%) | | | Attitude Score N (%) | | | Practices Score N (%) | | |
|---|---|---|---|---|---|---|---|---|---|---|
| | | Good = 153 (52.2) | Poor = 140 (47.8) | P value | Positive = 139 (47.4) | Negative = 154 (52.6) | P value | Good = 22 (7.5) | Poor = 271 (92.5) | P value |
| **Age** | | | | | | | | | | |
| *18–24 years* | 30 (10.2) | 12 (40) | 18 (60) | 0.142 | 13 (43.3) | 17 (56.7) | 0.077 | 0 (0.0) | 30 (100) | **0.043** |
| *25–34 years* | 184 (62.8) | 97 (52.7) | 87 (47.3) | | 96 (52.2) | 88 (47.8) | | 12 (6.5) | 172 (93.5) | |
| *35–44 years* | 60 (20.5) | 30 (50.0) | 30 (50.0) | | 20 (33.3) | 40 (66.7) | | 6 (10.0) | 54 (90.0) | |
| *>45 years* | 19 (6.5) | 14 (73.7) | 5 (26.3) | | 10 (52.6) | 9 (47.4) | | 4 (21.1) | 15 (78.9) | |
| **Sex** | | | | | | | | | | |
| *Male* | 158 (53.9) | 79 (50.0) | 79 (50.0) | 0.411 | 75 (47.5) | 83 (52.5) | 0.992 | 11 (7.0) | 147 (93.0) | 0.701 |
| *Female* | 135 (46.1) | 74 (54.8) | 61 (45.2) | | 64 (47.4) | 71 (52.6) | | 11 (8.1) | 124 (91.9) | |
| **Location** | | | | | | | | | | |
| *Urban* | 219 (74.7) | 113 (51.6) | 106 (48.4) | 0.715 | 106 (48.4) | 113 (51.6) | 0.571 | 15 (6.9) | 204 (93.1) | 0.461 |
| *Rural* | 74 (25.3) | 40 (54.0) | 34 (46.0) | | 33 (44.6) | 41 (55.4) | | 7 (9.5) | 67 (90.5) | |
| **Education level** | | | | | | | | | | |
| *Diploma* | 133 (45.4) | 77 (57.9) | 56 (42.1) | 0.161 | 73 (54.9) | 60 (45.1) | 0.054 | 12 (9.0) | 121 (91.0) | 0.411 |
| *Degree* | 132 (45.0) | 61 (46.2) | 71 (53.8) | | 56 (42.4) | 76 (57.6) | | 7 (5.3) | 125 (94.7) | |
| *Masters* | 28 (9.6) | 15 (53.6) | 13 (46.4) | | 10 (35.7) | 18 (64.3) | | 3 (10.7) | 25 (89.3) | |
| **Years of experience** | | | | | | | | | | |
| *<1 year* | 30 (10.2) | 13 (43.3) | 17 (56.7) | 0.311 | 16 (53.3) | 14 (46.7) | 0.266 | 1 (3.3) | 29 (96.7) | **0.045** |
| *1–5 years* | 97 (33.1) | 46 (47.4) | 51 (52.6) | | 46 (47.4) | 51 (52.6) | | 7 (7.2) | 90 (92.8) | |
| *6–10 years* | 113 (38.6) | 62 (54.9) | 51 (45.1) | | 58 (51.3) | 55 (48.7) | | 5 (4.4) | 108 (95.6) | |
| *>11 years* | 53 (18.1) | 32 (60.4) | 21 (39.6) | | 34 (64.1) | 19 (35.9) | | 9 (17.0) | 44 (83.0) | |
| **Workplace Setting** | | | | | | | | | | |
| *District/ Provincial Health office* | 7 (2.4) | 7 (100) | 0 (0.0) | 0.127 | 2 (28.6) | 5 (71.4) | 0.168 | 4 (57.1) | 3 (42.9) | **0.003** |
| *Health Centre/ Clinic* | 46 (15.7) | 23 (50.0) | 23 (50.0) | | 28 (60.9) | 18 (39.1) | | 3 (6.5) | 43 (93.5) | |
| *Private Hospital* | 18 (6.1) | 10 (55.6) | 8 (44.4) | | 6 (33.3) | 12 (66.7) | | 1 (5.6) | 17 (94.4) | |
| *Public Hospital* | 194 (66.2) | 99 (51.0) | 95 (49.0) | | 92 (47.4) | 102 (52.6) | | 11 (5.7) | 183 (94.3) | |
| *Research or Academic institution* | 28 (9.6) | 14 (50.0) | 14 (50.0) | | 11 (39.3) | 17 (60.7) | | 3 (10.7) | 25 (89.3) | |
| **Province of Practice** | | | | | | | | | | |
| *Central Province* | 21 (7.2) | 8 (38.1) | 13 (61.9) | 0.863 | 11 (52.4) | 10 (47.6) | 0.329 | 5 (23.8) | 16 (76.2) | 0.086 |
| *Copperbelt Province* | 115 (39.2) | 60 (52.2) | 55 (47.8) | | 56 (48.7) | 59 (51.3) | | 8 (7.0) | 107 (93.0) | |
| *Eastern Province* | 18 (6.1) | 10 (55.6) | 8 (44.4) | | 7 (38.9) | 11 (61.1) | | 1 (5.6) | 17 (94.4) | |
| *Luapula Province* | 12 (4.1) | 7 (58.3) | 5 (41.7) | | 4 (33.3) | 8 (66.7) | | 0 (0) | 12 (100) | |
| *Lusaka Province* | 33 (11.3) | 17 (51.5) | 16 (48.5) | | 17 (51.5) | 16 (48.5) | | 1 (3.0) | 32 (97.0) | |
| *Muchinga Province* | 9 (3.1) | 5 (55.6) | 4 (44.4) | | 4 (44.4) | 5 (55.6) | | 1 (11.1) | 8 (88.9) | |
| *North-Western Province* | 15 (5.1) | 9 (60.0) | 6 (40.0) | | 3 (20.0) | 12 (80.0) | | 3 (20.0) | 12 (80.0) | |
| *Northern Province* | 6 (2.1) | 4 (66.7) | 2 (33.3) | | 5 (83.3) | 1 (16.7) | | 0 (0) | 6 (100.0) | |

*(Continued)*

**Table 1.** (Continued)

| Variable | Frequency (%) | Knowledge Score N (%) | | | Attitude Score N (%) | | | Practices Score N (%) | | |
|---|---|---|---|---|---|---|---|---|---|---|
| | | Good = 153 (52.2) | Poor = 140 (47.8) | P value | Positive = 139 (47.4) | Negative = 154 (52.6) | P value | Good = 22 (7.5) | Poor = 271 (92.5) | P value |
| *Southern Province* | 56 (19.1) | 27 (48.2) | 29 (51.8) | | 29 (51.8) | 27 (48.2) | | 2 (3.6) | 54 (96.4) | |
| *Western Province* | 8 (2.7) | 6 (75.0) | 2 (25.0) | | 3 (37.5) | 5 (62.5) | | 1 (12.5) | 7 (87.5) | |

Data are presented as frequency (percentage). Knowledge, attitude, and practice outcomes were dichotomized into good/poor or positive/negative based on predetermined cut-off scores. P-values were obtained using the Chi-square test of independence; Fisher's exact test was applied where expected cell counts were < 5; Bold represented statistical significance which was set at p < 0.05. N = number of respondents. % = percentage.

knowledge scores of 33.3% in Jordan [28], 36.5% in Indonesia [29], 38.5% in Ethiopia [30] and 48% in Saudi Arabia [31]. These variations may be attributed to differences in study settings, sample size, timeframes, perceptions of Mpox infection, sources of information, and whether participants had received relevant training, all of which could contribute to the observed discrepancies.

Our study revealed that 47.4% of laboratory professionals exhibited a positive attitude towards Mpox. This is lower than the 51.7% reported across 27 countries [26], 62% reported in Ethiopia [30] and 85% observed in Saudi Arabia [27]. This difference is likely to be attributed to variations in sample size, study design, geographical context, and the categories of healthcare professionals included in each study.

In our survey, Mpox preventive practices included regular participation in training sessions and workshops focused on emerging infectious diseases, as well as the consistent use of personal protective equipment such as gloves, gowns, masks, and eye protection when managing patients with suspected infections [32]. Our survey revealed a 68.3% of respondents actively keeping themselves informed on current public health threats through credible sources and demonstrated awareness of the early signs and symptoms of Mpox. In Mpox suspected cases, 26.6% LPs indicated that they would promptly isolate the patient and report the case to relevant public health authorities in accordance with established guidelines. Furthermore, 94.5% reported strict adherence to infection prevention and control protocols, including proper hand hygiene, environmental sanitation, and biomedical waste management. Finally, 26.6% of HCPs expressed confidence in their ability to respond to a potential Mpox outbreak within their facilities, citing the presence of outbreak preparedness plans and routine staff training. These findings highlight a generally high level of awareness and adherence to Mpox prevention practices among qualified LPs in the public health sector. Our study revealed an extremely poor overall level of Mpox preventive practices of 7.5%. This is lower than 51.8% reported in Vietnam [22].

Our study found that 72% of LPs expressed willingness or support for mandatory Mpox vaccination, a rate slightly similar with the 67.7% reported by Kumar et al. (2022) among HCPs in Pakistan [20]. However, our findings were higher than those from a global systematic review and meta-analysis (58%) [33] and the 58.6% acceptance rate documented by Ricco et al. (2022) in Italy [34]. These variations may reflect differences in study populations, regional vaccination policies, or levels of Mpox awareness at the time of data collection.

Our finding that 62.5% of participants could not correctly identify Mpox transmission modes is particularly concerning. This significant gap may be attributed to systemic issues such as limited access to structured training programs and updates on emerging pathogens. Furthermore, in the digital age, health professionals increasingly turn to online sources for information, which can be a double-edged sword, providing rapid updates but also potentially exposing them to misinformation or inconsistent messaging [35]. This underscores the need for institutional efforts to provide regular, evidence-based updates and perhaps even training on digital health literacy to ensure laboratory professionals can critically appraise information sources.

**Table 2. Healthcare professionals Responses to Knowledge, Attitude and Practices Questions about Monkeypox in Zambia.**

| | | N (%) | |
|---|---|---|---|
| | **Knowledge Questions** | **Correct Answers** | **Incorrect Answers** |
| Q1 | Have you heard of Monkeypox (Mpox) before? | 288 (98.3) | 5 (1.7) |
| Q2 | What is the causative agent of Monkeypox? | 290 (99.0) | 3 (1.0) |
| Q3 | What are the common modes of Monkeypox transmission? (Select all that apply) | 110 (37.5) | 183 (62.5) |
| Q4 | What are the common symptoms of Monkeypox? (Select all that apply) | 110 (37.5) | 183 (62.5) |
| Q5 | Is Monkeypox a zoonotic disease? | 278 (94.9) | 15 (5.1) |
| Q6 | What is the incubation period of Monkeypox? | 182 (62.1) | 111 (37.9) |
| Q7 | Is there an effective vaccine available for Monkeypox? | 135 (46.1) | 158 (53.9) |
| Q8 | What are the potential complications of Monkeypox? (Select all that apply) | 70 (23.9) | 223 (76.1) |
| Q9 | Can Monkeypox be diagnosed through laboratory testing? | 290 (99.3) | 2 (0.7) |
| Q10 | Which of the following groups are considered at high risk for severe complications from mpox? (Select all that apply.) | 186 (63.5) | 107 (36.5) |
| Q11 | Which of the following sample types is most appropriate for the diagnosis of Mpox? | 233 (79.5) | 60 (20.5) |
| Q12 | Which of the following is the most appropriate laboratory test for confirming a Mpox diagnosis? | 250 (85.9) | 41 (14.1) |
| | **Attitude Questions** | | |
| Q1 | Do you believe Monkeypox is a serious public health threat in Zambia? | 268 (91.5) | 25 (8.5) |
| Q2 | How concerned are you about the risk of Monkeypox infection in healthcare settings? | 236 (80.6) | 57 (19.4) |
| Q3 | Do you think healthcare professionals should receive specialized training on Monkeypox management? | 288 (98.3) | 5 (1.7) |
| Q4 | Do you believe the Hospitals are adequately prepared to handle a Monkeypox outbreak? | 232 (79.2) | 61 (20.8) |
| Q5 | How confident are you in your ability to diagnose and manage Monkeypox cases? | 57 (19.5) | 236 (80.5) |
| Q6 | Do you support the mandatory vaccination of healthcare workers against Monkeypox if a vaccine is available? | 211 (72.0) | 82 (28.0) |
| Q7 | Do you think people with Monkeypox should be isolated to prevent the spread of the disease? | 272 (92.8) | 21 (7.2) |
| Q8 | How do you feel about the stigma associated with Monkeypox? | 182 (62.1) | 111 (37.9) |
| Q9 | Do you believe Monkeypox can be effectively controlled through public health interventions? | 284 (96.9) | 9 (3.1) |
| Q10 | Should Monkeypox surveillance and reporting systems be strengthened in Zambia? | 287 (98.0) | 6 (2.0) |
| | **Practices Questions** | | |
| Q1 | Have you ever attended a training or workshop on Monkeypox? | 18 (6.1) | 275 (93.9) |
| Q2 | How often do you use personal protective equipment (PPE) when handling patients with suspected infectious diseases? | 258 (93.1) | 19 (6.9) |
| Q3 | Do you regularly update yourself on emerging infectious diseases like Monkeypox? | 200 (68.3) | 93 (31.7) |
| Q4 | Have you ever encountered a suspected Monkeypox case in your healthcare facility? | 83 (29.3) | 200 (70.7) |
| Q5 | If you suspect a Monkeypox case, what would be your first step? | 78 (26.6) | 215 (73.4) |
| Q6 | Do you follow infection prevention and control (IPC) guidelines when managing patients with infectious diseases? | 259 (94.5) | 15 (5.5) |
| Q7 | How frequently do you educate patients and colleagues about Monkeypox prevention? | 33 (11.3) | 260 (88.7) |
| Q8 | Have you ever participated in Monkeypox surveillance or research? | 18 (6.1) | 275 (93.9) |
| Q9 | Do you know the reporting procedure if you identify a suspected Monkeypox case? | 88 (30.0) | 205 (70.0) |
| Q10 | If a Monkeypox outbreak occurs in your facility, do you feel prepared to handle cases? | 78 (26.6) | 215 (73.4) |

In the multivariate logistic regression analysis workplace setting a significant predictors of Mpox-related practices among LPs. LPs working in health centres/clinics, private hospitals, and research or academic institutions were significantly less likely to report good Mpox-related practices compared to those working in district or provincial health offices.

**Table 3. Factors associated with practices in logistic regression.**

| Variable | Univariate analysis | | Multivariate analysis | |
|---|---|---|---|---|
| | OR (95% CI) | P value | AOR (95% CI) | P value |
| **Age** | | | | |
| *>45 years* | Ref | | Ref | |
| *18–24 years* | (Empty) | (Empty) | (Empty) | (Empty) |
| *25–34 years* | 0.26 (0.07, 0.91) | **0.035** | 0.95 (0.099, 9.27) | 0.971 |
| *35–44 years* | 0.41 (0.10, 1.67) | 0.217 | 1.01 (0.19, 5.40) | 0.985 |
| **Years of experience** | | | | |
| *<1 year* | Ref | | Ref | |
| *1–5 years* | 2.25 (0.26, 19.1) | 0.456 | 1.02 (0.10, 9.86) | 0.981 |
| *6–10 years* | 1.34 (0.15, 11.9) | 0.792 | 0.46 (0.04, 4.88) | 0.522 |
| *>11 years* | 5.93 (0.71, 49.3) | 0.100 | 1.42 (0.08, 23.4) | 0.804 |
| **Workplace Setting** | | | | |
| *District/ Provincial Health office* | Ref | | Ref | |
| *Health Centre/Clinic* | 0.05 (0.007, 0.34) | **0.002** | 0.07 (0.009, 0.54) | **0.011** |
| *Private Hospital* | 0.04 (0.003, 0.54) | **0.015** | 0.05 (0.003, 0.74) | **0.030** |
| *Public Hospital* | 0.04 (0.008, 0.22) | **<0.0001** | 0.06 (0.011, 0.39) | **0.003** |
| *Research or Academic institution* | 0.09 (0.01, 0.61) | **0.014** | 0.10 (0.01, 0.77) | **0.027** |

OR = Odds Ratio; AOR = Adjusted Odds Ratio; CI = Confidence Interval; Ref = Reference category. Variables with p < 0.05 in the multivariate analysis were considered statistically significant.

This may be due to reduced access to formal training, and updated guidelines, which can limit opportunities for LPs to adopt and maintain optimal Mpox-related practices.

### Strengths and limitations

This study has several limitations that should be acknowledged. First, the survey was conducted in English, which may have introduced a selection bias, as only participants with sufficient English proficiency could complete the questionnaire. This requirement may imply a higher formative capacity among respondents, potentially overestimating knowledge and attitudes compared to the broader population of laboratory professionals. Secondly, we did not assess participants' sources of information. The influence of online health content and misinformation was not captured, yet digital platforms are a primary source of information and can significantly affect Mpox-related knowledge and practices through exposure to misinformation and variable quality content [35]. Thirdly, the exclusive use of an online survey format inherently excluded LPs lacking reliable internet access, likely underrepresenting practitioners in rural or resource-constrained settings. Finally, the reliance on self-reported data may have introduced social desirability and recall biases. Despite these limitations, the study possesses several notable strengths. To the best of our knowledge, this is the first study in Zambia to assess laboratory professionals' KAP regarding Mpox. The use of a structured questionnaire enhanced reliability, and the diverse provincial coverage of participants adds to the generalizability of the findings within the Zambian laboratory professional community.

Despite these limitations, the study possesses several notable strengths. To the best of our knowledge, this is the first study in Zambia to assess laboratory professionals' KAP regarding Mpox. As such, it provides valuable baseline data that can guide the development of targeted training initiatives, awareness campaigns, and evidence-informed policy interventions. The use of a structured and standardized questionnaire enabled consistent data collection across participants, enhancing the reliability of the findings. Furthermore, the inclusion of a diverse sample of LPs from multiple provinces

contributes to the breadth and contextual relevance of the results, offering a more comprehensive understanding of the current KAP landscape among laboratory workers in Zambia.

Importantly, these findings offer a broader reflection of the country's overall readiness or lack thereof, to respond to emerging infectious disease outbreaks. The study highlights significant gaps in outbreak preparedness, underscoring the urgent need for coordinated action by the Ministry of Health and key stakeholders. To strengthen the resilience of the health system, we propose priority interventions: including investment in health education, surveillance systems, and the development of responsive and proactive policies. Such measures are essential to ensure that healthcare professionals are adequately equipped to manage and mitigate the impact of future outbreaks like Mpox.

## Conclusion

We assessed the knowledge, attitudes, and practices related to Mpox among laboratory professionals in Zambia, identifying key gaps in awareness and behavioural practices. The findings revealed a low knowledge level, a low positive attitude, and poor practices towards Mpox. These insights are important for informing targeted education and training programs aimed at improving Mpox-related preparedness and response among healthcare workers. By addressing the identified gaps, Zambia can strengthen its capacity to prevent, detect, and mitigate the spread and impact of Mpox, especially in the face of emerging global health threats. This study provides foundational evidence to support policy development, resource allocation, and capacity-building initiatives to enhance outbreak preparedness and public health resilience in Zambia.

## Supporting information

**S1 File. Questionnaire form.**
(PDF)

**S2 File. Data.**
(XLSX)

**S3 File. Strobe checklist.**
(DOCX)

## Author contributions

**Data curation:** David Chisompola, John Nzobokela.

**Formal analysis:** David Chisompola, John Nzobokela.

**Investigation:** David Chisompola, John Nzobokela, Elijah Chinyante, Nanjela Chidima, Allen Chipipa, Charlotte Nyirenda, Edward Phiri, Lucky Kalyapu, Sepiso K. Masenga.

**Methodology:** David Chisompola, John Nzobokela, Elijah Chinyante.

**Project administration:** David Chisompola.

**Supervision:** David Chisompola, Sepiso K. Masenga.

**Validation:** David Chisompola.

**Visualization:** David Chisompola, John Nzobokela.

**Writing – original draft:** David Chisompola.

**Writing – review & editing:** David Chisompola, John Nzobokela, Elijah Chinyante, Nanjela Chidima, Allen Chipipa, Charlotte Nyirenda, Edward Phiri, Lucky Kalyapu, Sepiso K. Masenga.

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
