## [Decision Letter · Decision Letter 0]

29 Jul 2025

Dear Dr. Chisompola,

Thank you for submitting your manuscript to PLOS ONE. After careful consideration, we feel that it has merit but does not fully meet PLOS ONE’s publication criteria as it currently stands. Therefore, we invite you to submit a revised version of the manuscript that addresses the points raised during the review process.

We look forward to receiving your revised manuscript.

Kind regards,

Muhammad Abbas Abid, MD

Academic Editor

PLOS ONE

Satff Editor comments: Please correct the term monkeypox into Mpox in the title.

Journal Requirements:

I3. f the reviewer comments include a recommendation to cite specific previously published works, please review and evaluate these publications to determine whether they are relevant and should be cited. There is no requirement to cite these works unless the editor has indicated otherwise. 

Additional Editor Comments:

Even though it is a KAP study, the study is important as knowledge about mpox is still limited. Yet, there are a number of changes required before it becomes a holistic and interesting read.

The study results cannot be generalized to all healthcare professionals as most of the included participants technologists.

The limitations section should be added upon and the reviewer comments incorporated to make it a more comprehensive read.

Good luck to the authors!

Reviewers' comments:

Reviewer's Responses to Questions

**Comments to the Author**

1. Is the manuscript technically sound, and do the data support the conclusions?

Reviewer #1: Partly

Reviewer #2: Yes

2. Has the statistical analysis been performed appropriately and rigorously?

Reviewer #1: No

Reviewer #2: Yes

3. Have the authors made all data underlying the findings in their manuscript fully available?

Reviewer #1: Yes

Reviewer #2: Yes

4. Is the manuscript presented in an intelligible fashion and written in standard English?

Reviewer #1: Yes

Reviewer #2: Yes

Reviewer #1: 1. The manuscript labels 46.5% knowledge accuracy as “moderate” without establishing any theoretical or empirical justification for that classification. There is no engagement with literature that defines meaningful thresholds for knowledge adequacy in healthcare preparedness contexts. This subjective categorization reduces the interpretive clarity of the findings. The authors should support their classifications using references to similar KAP studies or global health benchmarks to ensure their descriptors have an objective basis and allow readers to assess the severity of the observed gaps with greater precision.

2. The finding that 76.4% of participants could not correctly identify Mpox transmission modes is significant, yet the paper does not probe the reasons for this widespread misconception. There is no exploration of the sources participants rely on for infectious disease information or whether misinformation might play a role. As a result, the implications of this gap are left underdeveloped. The discussion could be improved by analyzing potential systemic contributors to poor knowledge, such as lack of access to continuing education or the influence of low-quality digital health content, especially in low-resource environments.

3. The sample is heavily skewed, with 63.6% of respondents being laboratory technologists or scientists. Despite this imbalance, the authors generalize their findings across all healthcare professionals in Zambia. Such overrepresentation introduces substantial bias, especially in measuring practices and attitudes, as lab personnel may differ in their patient interaction frequency. A more appropriate approach would involve stratified results or a weighting mechanism to ensure that the findings reflect Zambia’s broader healthcare workforce demographics more accurately.

4. A key limitation of this study is its omission of the influence that online health content and misinformation may have on healthcare professionals' Mpox-related knowledge and practices. The findings are discussed in isolation from the digital information environment that increasingly shapes professional awareness and response to emerging diseases. This narrows the scope of the analysis and limits its utility for training design. To address this, it is suggested that the authors review recent works such as https://doi.org/10.1109/CCWC62904.2025.10903713 and https://doi.org/10.1109/AiDAS63860.2024.10730443, which show how exposure to sentiment and toxicity in online platforms affects health discourse, and can inform how preparedness programs should adapt to the realities of information ecosystems.

5. There are unresolved concerns about participant anonymity and data integrity. The paper claims that duplicate responses were prevented via email verification, yet also states that all data were collected anonymously. This contradiction raises questions about how the research balanced verification with participant privacy, especially given the use of a Google Form. The authors need to clarify the process, explicitly stating how identity verification was managed without violating ethical principles of anonymity, particularly in light of the sensitive nature of self-reported practices.

Reviewer #2: The study is well formatted and methodologically sound and provides interesting insights into the levels of preparedness of health workers in Zambia, so that training resources can be improved.

MAJOR COMMENTS:

1. Line 52: The mode of transmission described as ‘direct transmission’ is unclear, it would be better described as something along the following lines: ‘direct contact with a person with the virus’, and maybe with the addition that this occurs through skin lesions, body fluids....

2. Discussion: You could introduce more commentary on the results than simply comparing data from other studies but try to give some kind of explanation for the findings based on the literature.

3. Strengths and limitations: Mention a possible bias of selection related to people responding, with the language, cause participants should understand English and this could imply a greater formative capacity and overestimate some values.

MINOR COMMENTS:

1. Line 15: error in the correspondent's e-mail address (p.2): Should correct “gmai.com” to “gmail.com.”

2. Line 50: Orthopoxvirus, all together, instead Ortho poxvirus

**Do you want your identity to be public for this peer review?** For information about this choice, including consent withdrawal, please see our Privacy Policy

Reviewer #1: No

Reviewer #2: **Yes: ** Inés Armenteros-Yeguas

---

## [Author Response · Author response to Decision Letter 1]

15 Sep 2025

Response to Reviewers

We thank the Editor for handling our manuscript (PONE-D-25-30146), titled "Knowledge, Attitudes, and Practices Toward Mpox Among Laboratory Professionals in Zambia: A Cross-Sectional Study," and the Reviewers for their insightful comments. Their feedback has been invaluable in significantly improving our manuscript. We have refined our analysis to focus exclusively on laboratory professionals and have incorporated new discussions and limitations as suggested. We believe these revisions have substantially strengthened the manuscript.

We have provided our point-to-point answers to the reviewers’ comments below.

Editor and Reviewers comments

Staff Editor comments: Please correct the term monkeypox into Mpox in the title.

Response: We thank the editorial team for this comment. We have updated the title as requested. It now reads “Knowledge, Attitudes, and Practices Toward Mpox Among Laboratory Professionals in Zambia: A Cross-Sectional Study”.

Additional Editor Comments:

Even though it is a KAP study, the study is important as knowledge about mpox is still limited. Yet, there are a number of changes required before it becomes a holistic and interesting read.

The study results cannot be generalized to all healthcare professionals as most of the included participants technologists.

Response: We thank the editorial team for this critical observation. We agree with this limitation. Consequently, we have revised the entire manuscript to focus solely on laboratory professionals (technologists and scientists). All analyses, results, and discussions now specifically pertain to this group, and we have adjusted the title, abstract, introduction, methods, and discussion to reflect this focused scope. Generalizations to other healthcare professionals have been removed.

The limitations section should be added upon and the reviewer comments incorporated to make it a more comprehensive read.

Response: We thank the editorial team for this suggestion. We have substantially expanded the 'Strengths and Limitations' section to incorporate all points raised by the reviewers, including the potential for selection bias due to the English-language survey, the influence of online information and misinformation, and the limitations of the sampling method.

Reviewer #1:

1. The manuscript labels 46.5% knowledge accuracy as “moderate” without establishing any theoretical or empirical justification for that classification. There is no engagement with literature that defines meaningful thresholds for knowledge adequacy in healthcare preparedness contexts. This subjective categorization reduces the interpretive clarity of the findings. The authors should support their classifications using references to similar KAP studies or global health benchmarks to ensure their descriptors have an objective basis and allow readers to assess the severity of the observed gaps with greater precision.

Response: We thank the reviewer for this crucial feedback. We have revised our interpretation to use the standard Bloom's cut-off point, which categorizes a score of ≤59% as low, 60-79% as moderate, and 80-100% as high. Our mean knowledge score of 52.2% therefore falls into the 'low' category, not 'moderate'. We have corrected this throughout the manuscript (Abstract, Results, Discussion) and have cited the use of this established benchmark (Ref 25: Halboup et al., 2023) to provide an objective basis for our interpretation.

2. The finding that 76.4% of participants could not correctly identify Mpox transmission modes is significant, yet the paper does not probe the reasons for this widespread misconception. There is no exploration of the sources participants rely on for infectious disease information or whether misinformation might play a role. As a result, the implications of this gap are left underdeveloped. The discussion could be improved by analyzing potential systemic contributors to poor knowledge, such as lack of access to continuing education or the influence of low-quality digital health content, especially in low-resource environments.

Response: We agree with the reviewer that this is a critical point. We have revised the Discussion section to delve deeper into the potential reasons for these significant knowledge gaps. We now explicitly mention the likely role of limited access to structured continuing education and the potential reliance on informal digital sources, which may expose professionals to misinformation. We have also expanded the Limitations section to formally acknowledge that our study did not assess information sources, which is a key area for future research.

3. The sample is heavily skewed, with 63.6% of respondents being laboratory technologists or scientists. Despite this imbalance, the authors generalize their findings across all healthcare professionals in Zambia. Such overrepresentation introduces substantial bias, especially in measuring practices and attitudes, as lab personnel may differ in their patient interaction frequency. A more appropriate approach would involve stratified results or a weighting mechanism to ensure that the findings reflect Zambia’s broader healthcare workforce demographics more accurately.

Response: We thank the reviewer for this important critique. We fully agree. Rather than stratifying or weighting the existing data, which was not designed for this purpose, we have chosen to refine the focus of our entire paper. We have revised the manuscript to specifically study and report on Laboratory Professionals only. The title, objectives, participant descriptions, results, and discussion have been updated accordingly. This provides a more accurate and meaningful contribution to the literature on a specific, critical subgroup of healthcare workers.

4. A key limitation of this study is its omission of the influence that online health content and misinformation may have on healthcare professionals' Mpox-related knowledge and practices. The findings are discussed in isolation from the digital information environment that increasingly shapes professional awareness and response to emerging diseases. This narrows the scope of the analysis and limits its utility for training design. To address this, it is suggested that the authors review recent works such as https://doi.org/10.1109/CCWC62904.2025.10903713 and https://doi.org/10.1109/AiDAS63860.2024.10730443, which show how exposure to sentiment and toxicity in online platforms affects health discourse, and can inform how preparedness programs should adapt to the realities of information ecosystems.

Response: We thank the reviewer for this insightful suggestion and for the relevant references. We have reviewed the suggested literature and have incorporated this perspective into our revised manuscript. The Limitations section now explicitly discusses this gap in our study and the potential influence of the digital information ecosystem. Furthermore, we have added a sentence to the Discussion suggesting that future training programs should include digital literacy components to help professionals navigate online information critically. The suggested references have been added to the reference list (Ref 35).

5. There are unresolved concerns about participant anonymity and data integrity. The paper claims that duplicate responses were prevented via email verification, yet also states that all data were collected anonymously. This contradiction raises questions about how the research balanced verification with participant privacy, especially given the use of a Google Form. The authors need to clarify the process, explicitly stating how identity verification was managed without violating ethical principles of anonymity, particularly in light of the sensitive nature of self-reported practices.

Response: We thank the reviewer for highlighting this contradiction and sincerely apologize for the oversight. To clarify: no personal identifying information, including email addresses, was collected. The Google Form was configured to use its built-in "Limit to 1 response" feature, which uses the respondent's Google account to prevent duplicates without transmitting the email address to the collector. This ensures anonymity. We have revised the Methods section to state this clearly: “The Google Form was configured to accept only one response per participant using its built-in “limit to one response” feature, which prevents duplicate responses without collecting identifying information. No email addresses or other personal identifiers were collected, ensuring complete participant anonymity.”

Reviewer #2:

The study is well formatted and methodologically sound and provides interesting insights into the levels of preparedness of health workers in Zambia, so that training resources can be improved.

MAJOR COMMENTS:

1. Line 52: The mode of transmission described as ‘direct transmission’ is unclear, it would be better described as something along the following lines: ‘direct contact with a person with the virus’, and maybe with the addition that this occurs through skin lesions, body fluids....

Response: We thank the reviewer for this suggestion to improve clarity. We have revised the sentence in the Introduction to read: “Human infections occur through animal spillover or direct contact with lesions, body fluids, or respiratory droplets of an infected person…”

2. Discussion: You could introduce more commentary on the results than simply comparing data from other studies but try to give some kind of explanation for the findings based on the literature.

Response: We thank the reviewer for this suggestion to deepen our discussion. We have significantly revised the Discussion section. Beyond comparing rates, we now provide potential explanations for our findings, particularly the low knowledge scores and the significant differences in practices based on workplace setting. We discuss factors such as access to training, the potential impact of information sources, and the specific roles of laboratory professionals in different settings.

3. Strengths and limitations: Mention a possible bias of selection related to people responding, with the language, cause participants should understand English and this could imply a greater formative capacity and overestimate some values.

Response: We thank the reviewer for this excellent point. We have added this specific limitation to the 'Strengths and Limitations' section: “First, the survey was conducted in English, which may have introduced a selection bias, as only participants with sufficient English proficiency could complete the questionnaire. This may have excluded laboratory professionals with lower English literacy, potentially leading to an overestimation of knowledge and attitude scores in our sample.”

MINOR COMMENTS:

1. Line 15: error in the correspondent's e-mail address (p.2): Should correct “gmai.com” to “gmail.com.”

Response: We thank the reviewer for spotting this error. We have corrected the corresponding author’s email address to d.chisompola@gmail.com

2. Line 50: Orthopoxvirus, all together, instead Ortho poxvirus

Response: We thank the reviewer for this correction. We have revised "Ortho poxvirus" to "Orthopoxvirus" in the Introduction.

Please include captions for your Supporting Information files at the end of your manuscript, and update any in-text citations to match accordingly. Please see our Supporting Information guidelines for more information: http://journals.plos.org/plosone/s/supporting-information.

Response: Thank you spotting this issue. We have added Supplementary files, S1. Questionnaire form, S2. Data and S3. Strobe checklist

---

## [Editor Report · Decision Letter 1]

10 Oct 2025

Knowledge, Attitudes, and Practices Toward Mpox Among Laboratory Professionals in Zambia: A Cross-Sectional Study

PONE-D-25-30146R1

Dear Dr. Chisompola,

We’re pleased to inform you that your manuscript has been judged scientifically suitable for publication and will be formally accepted for publication once it meets all outstanding technical requirements.

Kind regards,

Muhammad Abbas Abid, MD

Academic Editor

PLOS ONE
---

## [Editor Report · Acceptance letter]

PONE-D-25-30146R1

PLOS ONE

Dear Dr. Chisompola,

I'm pleased to inform you that your manuscript has been deemed suitable for publication in PLOS ONE. Congratulations! Your manuscript is now being handed over to our production team.

Kind regards,

on behalf of

Dr. Muhammad Abbas Abid

Academic Editor

PLOS ONE